# Serotonin—Its Synthesis and Roles in the Healthy and the Critically Ill

**DOI:** 10.3390/ijms22094837

**Published:** 2021-05-03

**Authors:** Marcela Kanova, Pavel Kohout

**Affiliations:** 1Department of Anaesthesiology and Intensive Care Medicine, University Hospital Ostrava, 70852 Ostrava-Poruba, Czech Republic; 2Institute of Physiology and Pathophysiology, Faculty of Medicine, University of Ostrava, Syllabova 19, 70300 Ostrava-Vítkovice, Czech Republic; 3Department of Internal Medicine, 3rd Faculty of Medicine, Charles University Prague and Teaching Thomayer Hospital, 14059 Prague, Czech Republic; pavel.kohout@ftn.cz

**Keywords:** serotonin, neurotransmitter, peripheral hormone, intestinal motility, energy metabolism, immunoregulatory functions, serotonin syndrome, critically ill

## Abstract

Serotonin (5-hydroxytryptamine, 5-HT) plays two important roles in humans—one central and the other peripheral—depending on the location of the 5-HT pools of on either side of the blood-brain barrier. In the central nervous system it acts as a neurotransmitter, controlling such brain functions as autonomic neural activity, stress response, body temperature, sleep, mood and appetite. This role is very important in intensive care, as in critically ill patients multiple serotoninergic agents like opioids, antiemetics and antidepressants are frequently used. High serotonin levels lead to altered mental status, deliria, rigidity and myoclonus, together recognized as serotonin syndrome. In its role as a peripheral hormone, serotonin is unique in controlling the functions of several organs. In the gastrointestinal tract it is important for regulating motor and secretory functions. Apart from intestinal motility, energy metabolism is regulated by both central and peripheral serotonin signaling. It also has fundamental effects on hemostasis, vascular tone, heart rate, respiratory drive, cell growth and immunity. Serotonin regulates almost all immune cells in response to inflammation, following the activation of platelets.

## 1. Introduction

This review gives a brief overview of serotonin synthesis followed by a summary of its signaling role both under normal physiological conditions and under critical illness. Serotonin carries out a number of immune functions as a neurotransmitter and as a peripheral hormone. It is critical for the inflammatory response, possibly influencing the development of the systemic inflammatory response syndrome (SIRS). Further, it affects cardiovascular and respiratory functions, controls platelet function and hemostasis. As an important gastrointestinal signaling molecule, serotonin has a number of diverse functions in metabolism, including influencing motor and sensory functions through its effects on the microbiome, as well as in controlling energy balance.

The aim of this review is to take the perspective of the intensive care specialist and highlight these roles of serotonin, from influencing immune functions, progression of sepsis and its influence on hemostasis. Another objective was to explore its effects on enteral nutrition, the risk of polypharmacy and the likelihood of developing the life-threatening serotonin syndrome.

## 2. The Role of Serotonin

### 2.1. Serotonin Synthesis

Serotonin (5-hydroxytryptamine, 5-HT) is located mainly in the serotoninergic neural network of the central nervous system, in the gastrointestinal (GI) tract and in platelets, where 5-HT is stored. Serotonin acts as both a neurotransmitter and as a peripheral hormone. However, most 5-HT production occurs in the enterochromaffin (EC) cells of the intestinal mucosa. The gut is the largest endocrine organ in human body, and it produces almost 95% of all the serotonin. This is triggered in response to acetylcholine, neuronal stimulation, increase in intraluminal gut pressure or low gut luminal pH. Serotonin is released into the lamina propria where dendritic cells, T lymphocytes and other immune cells are located, as well as into the intestinal lumen (Figure 1).

Peripheral serotonin synthesized by EC cells is taken up and stored in the platelets. Platelets take up 5-HT from the bloodstream through a serotonin reuptake transporter (SERT) on the membrane and store it in dense granules via the action of the vesicular monoamine transporter (VMAT-2). Platelets have the capacity to store and release 5-HT and thus regulate peripheral serotonin levels. Under normal conditions the blood serotonin level is maintained at 1–5 ng/mL, but it can increase 1000-fold upon platelet activation in response to inflammation [1,2,3,4].

Platelet-derived serotonin plays a role in everything that platelets do—in hemostasis, thrombosis and in immune functions. Extracellular serotonin originating from EC cells triggers an increase in the number of vesicles delivering SERT to the membrane, and thus 5-HT is sequestered in the platelet cells. The main role of platelets is regulation of hemostasis following injury. Once activated, platelets release serotonin, which is thus free to bind to receptors on various tissues. Serotonin binds to the 5-HT2A receptor on the platelets from which it is released, leading to platelet aggregation. 5-HT regulates expression of P- and E-selectins on endothelial cells and activates the rolling and adhesion of neutrophils. When serotonin is depleted, this innate immune reaction of neutrophils is reduced [1,5,6].

In the central nervous system, serotonin is produced by neurons located in the raphe nuclei in the brainstem. Serotonin regulates diverse behavioral manifestations such as mood, perception, memory and stress responses, as well as influencing parameters like circadian rhythms, body temperature and emesis. Nevertheless, most serotonin is located outside the CNS, where it modulates major organ functions such as heart rate, respiratory drive, vasoconstriction, intestinal motility and secretion. Serotonin also plays a critical role in endocrine secretion, in the control of energy balance and metabolism, as well as in the regulation of glucose homeostasis and lipid metabolism. 5-HT aids the regeneration of metabolic organs, e.g., liver regeneration following volume loss after resection [1].

A wide range of functions is made possible by the diversity of serotonin receptors. Fifteen different serotonin receptors belonging to seven receptor families (5-HTR1-7) are encoded by 18 genes (Table 1), and different cells express different serotonin receptors (Table 2). Almost all 5-HTR, except the 5HT3 group, are G- protein coupled receptors. The signaling pathway involves GTPase triggering a second messenger cascade, leading to protein modification (a process called serotonylation). Activation of 5-HTR1,5 reduces c-ATP, while 5-HTR4,6,7 activation increases c-ATP activity. The only exception is 5-HTR3, which is a nonselective cation channel [2].

Serotonin is synthetized in two steps from tryptophan, an essential amino acid acquired from food. The first and limiting enzymatic step is hydroxylation by tryptophan-hydroxylase (TPH) to produce 5-hydroxytryptophan. The TPH1 isoform is found in EC cells, and the TPH2 isoform is located in central and enteric neurons. The next step is conversion to 5-hydroxytryptamine by an aromatic L-amino acid decarboxylase. 5-HT is then packaged into vesicles by the vesicular monoamine transporter (VMAT). The sequestration of 5-HT to vesicles in the cytoplasm prevents rapid degradation by monoamine oxidase (MAO). MAO breaks down 5-HT to form 5-hydroxyindoleacetic acid (5-HIAA), which is mainly excreted in urine. The SERT relocates 5-HT into presynaptic terminals in the CNS, and into platelets (the main store of 5-HT). SERT is expressed in pulmonary and peripheral blood vessels, as well as in the GI tract [4].

Serotonin cannot cross the blood-brain barrier and, therefore, central and peripheral serotonin form two distinct pools. Serotonin synthesis depends on the level of circulating tryptophan absorbed from the diet. Tryptophan can cross the blood-brain barrier by the action of a cognate L-type amino acid transporter.

### 2.2. Immune Response

Peripheral serotonin is important for a proper immune response, especially in the fight against infection and sepsis in critically ill patients. 5-HT also impacts various inflammatory diseases, e.g., inflammatory conditions of the gut (inflammatory bowel disease), rheumatoid arthritis or allergic airway dispositions. Serotonin is a potent modulator of both the innate and the adaptive immune system through its binding to immune-cell 5-HT receptors [1].

The main source of 5-HT for immune cells and lymphoid tissue is platelets. Unlike mast cells, monocytes/macrophages, and T cells, which partly participate in serotonin production, platelets cannot produce 5-HT themselves. The serotonin they use is produced by EC cells, and they take up 5-HT from the blood stream by SERT action. After activation of platelets in response to a disrupted endothelium, the content of their vesicles is released. The innate immune system forms the first line of defense against invading pathogens and is nonspecific in its action; meaning that the same response is elicited by different stimuli, such as microorganisms, anaphylatoxins and other soluble factors. This diversity is due to the 5-HTRs on the surface of immune cells (neutrophils, eosinophils, monocytes/macrophages, dendritic cells, mast cells and natural killer cells). Platelets, mast cells and basophils release 5-HT in response to inflammatory substances like cytokines, the platelet activating factor (PAF) and the complement system. Serotonin acts as a chemotactic agent, increasing proinflammatory cytokine secretion (interleukins IL-1, IL-6, NFκB) and enhancing phagocytosis. 5-HT stimulation intensifies interferon-gamma production by natural killer cells (NK), stimulates cytotoxicity, and is important for antiviral activity. Those who take selective serotonin reuptake inhibitors (SSRIs) have higher numbers of natural killers (NK). Serotonin not only increases protection against infections, but also acts to counteract oxidative damage [2,3].

On the other hand, serotonin plays an important role in adaptive immunity. This defense is already selective, the specificity being determined due to antigen-specific recognition by T and B lymphocytes. This response is slower, is responsible for immunological memory, and is highly specific (due to antibody specificity). Serotonin stimulates antigen--presenting cells (dendritic cells, macrophages), that activate adaptive immune reactions in which 5-HTR7 plays a major role. Serotonin can directly affect both T and B lymphocytes through their 5-HTR [1,2,3,7,8].

Gut-derived serotonin can affect immunity not only locally (inflammatory bowel disease (IBD), celiac disease, irritable bowel syndrome (IBS)), but it can enter the blood stream and participate in systemic inflammation via its impact on immune cells [1]. 

Most of the serotonin produced by EC cells is released into the lamina propria where it activates immune cells. Serotonin modulates mutual interactions between the EC cells, neurons of the myenteric plexus and gut-associated lymphoreticular tissue (GALT). In patients with inflammatory bowel disease (IBD), we see downregulated SERT and increased THP1 expression. 5-HT increases the production of reactive oxygen species (ROS) through NADPH, and subsequently enhances cytokine production (IL6 & IL8), resulting in the adhesion of monocytes to GI epithelial cells [9]. This may also be potentiated by increased 5-HT signaling. 5-HT modulation is also associated with obesity, as this can be characterized as chronic low-grade inflammation [10].

Serotonin affects a wide variety of immune functions; it can affect systemic inflammation, as well as a number of autoimmune diseases. Serotonin modulation can, therefore, be potentially useful as a therapeutic strategy.

### 2.3. The Gut-Brain-Microbiome Axis

The connection between the gut and the brain allows their functions to be influenced by serotonin on both sides of the gut-brain axis. The importance of gut microbiota is increasingly being recognized, and 5-HT makes gut-brain-microbiome crosstalk possible through the microbial influence on tryptophan metabolism and serotonergic synthesis. The developing serotonergic system in early childhood may be vulnerable to microbial colonization prior to the formation of stable adult-like gut microbiota. In the elderly, the decreased number and diversity of gut bacteria can cause serotonin-related health problems. This may be explained by the ability of gut microbiota to control host tryptophan metabolism via the kynurenine pathway and to influence immune and stress responses of the host organism. Only a small portion of the 5-HT pool seems to be directly synthetized by gut bacteria such as *E. coli*, *Corynebacterium* spp. and *Streptococcus* spp. [11,12,13]. Although the gut microbiota is generally stable, it can be altered during enteric infections, stress response and antibiotic treatment. Neurodestructive processes that can lead to dementia and Alzheimer’s disease (AD) begin with gut dysbiosis, local and systemic inflammation and dysregulation of the gut-brain axis. Increased gut permeability results in invasion of different bacteria, viruses and their neuroactive products that support neuroinflammatory reactions in the brain [14].

The strong connection between activation of peripheral immune cells and CNS-located immune cells is responsible for the associations between inflammation, immune activation and neuropsychiatric disorders. That is why treating depressive disorders with selective serotonin reuptake inhibitors (SSRIs) can affect peripheral immune reactions [1]. SSRIs have an immunosuppressive effect by reducing peripheral immune cell proliferation, cytokine production and apoptosis modulation [15,16]. SSRIs can be used to treat not only mood disruption, but also autoimmune disorders such as inflammatory bowel disease (IBD). These drugs affect the role of peripheral serotonin in T-cell mediated gut inflammation. IBD is an autoimmune disease with excessive Th1 and Th17 responses. There is mounting evidence of a link between 5-HT and T-lymphocytes, suggesting that serotonin modulation may be useful for therapy [17,18].

The other neurotransmitter that affects immune reactions is acetylcholine. The vagus nerve (VN) is involved in neuro-immune and brain-gut connections. VN is a major part of the autonomic nervous system, forms the link between the central nervous system (CNS) and major visceral organs, and is the crossroads for neuro-immune interactions. Vagal efferents have an immunosuppressive effect, known as the cholinergic anti-inflammatory pathway. Stimulating the VN (through e.g., exercise, nutrition and parasympathomimetic therapies) can improve pathological conditions associated with autonomic nervous system (ANS) imbalance including IBD [19].

Thus, understanding the impact of 5-HT on inflammation and other neurotransmitters can potentially be useful for modulating gastrointestinal motor and sensory functions, especially in critically ill patients. It is particularly interesting given that in these patients, the intestinal microbiome, as well as gut-brain-microbiome crosstalk, are significantly affected (e.g., due to the effect of antibiotics, vasopressors and parenteral nutrition). 

### 2.4. Metabolism

Both peripheral and central serotonin signaling are necessary to maintain energy balance. Specific serotonin receptors modulate specific brain regions. Brainstem serotonin neurons have projections ascending to cortical, limbic, midbrain and hindbrain regions. Serotoninergic neurons modulate nearly every human behavior including appetite, mood and food intake, as well as affecting energy balance. Each behavior is regulated by multiple serotonin receptors in multiple brain regions [3]. The hypothalamus is pivotal for energy balance signals and, as this limbic area has a porous blood-brain barrier, it can sense a wide range of circulating nutrients and hormones. Thus, it is in an ideal position to sense afferent signals transmitted through the VN from the gastrointestinal tract and other visceral organs. The hypothalamus also receives important information from the olfactory cortex and other brainstem nuclei (the raphe nuclei). In addition, the hypothalamus seems to have two loci with opposing effects on behavior—the medially-located nuclei satiety center and the laterally-located starvation center. A lesion in the medial part leads to hyperphagia and obesity, whereas a lesion in the lateral part produces hypophagia [20].

Pharmacological or genetic manipulation of several serotonin receptors can induce either orexigenic effects leading to obesity, or its converse, i.e., they can be anorectic. 5-HTR2C is a receptor that has a proven anorectic effect. 5-HTR2C knock-out mice are typically obese. Perturbation of 5-HTR2C RNA causes morbid obesity due to hyperphagia (lack of satiety), together with cognitive impairment and a short stature in the Prader-Willi syndrome [21]. The 5-HTR2C receptor affects meal frequency, while the 5-HTR1B serotonin receptor in the arcuate nucleus affects meal duration. However, this role is not restricted to serotonin receptors alone. Proopiomelanocortin (POMC) and its receptors located in the arcuate nucleus also have a significant role in energy metabolism. While the primary function of this system is anorectic, the effect of another group of neurons in the arcuate nucleus is orexigenic, since they produce gamma-aminobutyric acid (GABA), neuropeptide Y (NPY) and agouti-related peptide (AgRP), which inhibit POMC. The extra-hypothalamic serotonin receptor 5-HTR6 in the striatum produces orexigenic behavior. However, the network of neurons and their interactions are highly complex and can have opposing effects [20,22,23]. 

The blood-brain barrier separates the central and the peripheral serotonin pools. Almost 95% of serotonin is synthetized by the EC with a minority coming from the serotoninergic neuron network. EC cells contain tryptophan hydroxylase-1 (TPH-1), whereas serotoninergic neurons contain TPH-2. The presence of food (resulting in mechanical pressure and acid) in the intestinal lumen stimulates the cholinergic VN in the enteric submucosa that subsequently activates serotonin release from EC cells. GI epithelium cells and EC cells are constantly being renewed, and so EC cells produce 5-HT in excess and release it into the connective tissue. There it acts locally in a paracrine fashion and can stimulate both extrinsic and intrinsic afferent neurons (submucosal and myenteric plexus; Figure 2). Serotonin action is terminated by reuptake from the place of action into enterocytes, where 5-HT is catabolized by intracellular monoamine oxidase (MAO).

Peripheral serotonin plays the critical role of a signaling molecule in enteric neurotransmission, initiating peristaltic and secretory reflexes. Serotonin regulates digestion partly by stimulating pancreatic enzyme secretion. It is also important for gut motility—5-HTR1 on submucosal neurons initiate peristalsis and 5-HTR4 maintains peristalsis. 5-HTR4 are presynaptic receptors on cholinergic nerve terminals. Primary transmission in peristaltic reflexes is cholinergic and 5-HTR4 stimulation amplifies the strength of synaptic signaling. 5-HTR4 agonists (cisapride, tegaserod) are prokinetics. 5-HTR3 receptors located on VN terminals (myenteric plexus) in the gastrointestinal tract are responsible for the nausea sensation, as serotonin stimulates these receptors in the vomiting center. 5-HTR3 antagonists (ondansetron, granisetron) are effective antiemetic drugs. These drugs do not interfere with the activation of submucosal neurons initiating peristalsis, so they are useful in treating complications following cancer chemotherapy [24]. 

Serotonin plays a central role in carcinoid syndrome by promoting intestinal motility in patients with neuroendocrine tumors. These tumors belong to a heterogeneous group of malignancies that originate in endocrine and other cells derived from the neural crest. Clinical signs vary depending on the peptide released, and can include diarrhea, flushing, mesenteric fibrosis and abdominal pain. The level of serotonin in plasma and that of its metabolite 5-hydroxyindoleacetic acid in urine are usually elevated. New inhibitors of peripheral serotonin synthesis appear to relieve some of the symptoms and improve quality of life. Nevertheless, much remains to be more thoroughly investigated [25].

In summary, serotonin causes smooth muscle contraction in response to a food bolus. Thus, from the point of view of intensive-care specialists, this role of serotonin is crucial in determining how to use enteral feeding in critically ill patients. Moreover, it is also necessary to account for tumors that produce serotonin.

### 2.5. Bolus vs. Continual Enteral Feeding

Metabolic changes in critically ill patients leads to the development of hypercatabolism, increased energy requirement, insulin resistance and a massive stimulation of proteolysis. Thus, the critically ill lose muscle mass as a consequence of these metabolic changes that ensure survival. Polyneuromyopathy in the critically ill is thus a major problem that exacerbates mortality and morbidity. These patients are mainly fed continuously as they tolerate it better, and it is easier to correct glycaemia by continuously giving insulin. But with regard to stimulating proteosynthesis (rapamycin m-TOR system) and suppressing proteolysis (proteasome-ubiquitin system), this method appears to be nonphysiological. Feeding boluses stimulate natural enterohormonal pathways with the release of bioactive intestinal polypeptides (cholecystokinin, peptide YY, glucagon-like peptide). The release of insulin and activation of the rapamycin system leads to the activation of proteosynthesis. On the other hand, the presence of energy-dense food in the small intestine activates the ileal brake. Critically ill patients experience frequent dietary intolerance due to this upregulation of enterogastric feedback. As pointed out before, once food enters the GI tract, intestinal serotonin regulates pancreatic secretion and peristaltic waves. Drugs targeting 5-HTR3 and 5-HTR4 have been used to treat irritable bowel syndrome. However, using their prokinetic and insulin stimulating potential to treat GI disorders in the critically ill (e.g., paralytic ileus after abdominal trauma or surgery), remains questionable [26,27,28].

Peripheral serotonin determines bowel movement—it is increased in diarrhea and decreased in constipation. Postprandial symptoms appear earlier and do not coincide with the peak of serotonin at about 2 to 3 h after meal. It is possible that the major actors are upper intestinal mediators such as gastrin, cholecystokinin, motilin, pancreatic polypeptide and secretin, or members of the vasoactive intestinal polypeptide family. There are also differences in platelet SERT function—reduced SERT causes higher circulating serotonin and diarrhea. The opposite is true for increased SERT in platelet membranes, resulting in higher reuptake and reduced serotonin levels in the bloodstream [29]. Altered motility causing hypoxia in patients with IBD was shown to induce serotonin synthesis by stimulating adenosine receptors, leading to diarrhea [30]. 

This highlights the complexity of the serotoninergic regulation of energy balance at multiple steps by different, and frequently opposing, mechanisms. Deeper understanding of the serotonin signaling system in the two (myenteric and submucosal) neural plexuses, and its influence on GI motility and sensitivity, has led to drug development for functional GI disorders. 5-HTR4 agonists have a stimulating effect on gut motility, enable faster gastric emptying, and faster bowel transit. 5-HTR4 agonists are used as prokinetics in chronic constipation and in constipation-predominant IBS. 5-HTR3 antagonists have an inhibitory effect on gut motility and secretion, slower small intestine bowel transit, decreased intestinal secretion and colonic tone, and have antiemetic activity. These drugs (setrons) alleviate serotonin-evoked discomfort, which induces nausea. They are also used for diarrhea-predominant IBS [26].

It is thus clear from the literature that serotonin and serotoninergic agents can be effective in treating chronic diarrhea and constipation. New topical agents/drugs are now available, particularly to manage functional gastrointestinal disorders. Further, bolus enteral feeding may also help alleviate skeletal muscle atrophy.

### 2.6. Serotonin and Anesthesia

The serotonin system seems to play an important role in modulating the sleep/wake cycle and can affect anesthesia in the same way, mainly inhalation anesthesia. Released 5-HT spreads through the ascending reticular formation, responsible for regulating sleep-wake transitions. Serotonin release is naturally reduced during slow wave sleep, and inhalation anesthetic agents suppress serotonin release. Thus, patients taking serotoninergic antidepressants may need higher doses of such anesthetics [31]. Serotonin also modulates pain control. It is released from peripheral nerve fibers in locally inflamed tissue, and nociceptive information is carried by nerve fibers to the CNS. Serotoninergic brainstem neurons first affect descending projections into the spinal cord and thereby modulate nociception. Moreover, raphe serotoninergic neurons have ascending projections into cortical and limbic areas, and thus affect the psychological perception of pain [32,33].

### 2.7. Serotonin Syndrome

Serotonin syndrome (SS) is a potentially life-threatening condition, which is due to excessive serotonin action on the central and peripheral nervous system. SS is an adverse drug reaction due to serotoninergic medication overdose, mostly via inadvertent interaction of several serotoninergic drugs. Especially from the point of view of intensive care specialists, this condition is not as rare as previously thought. SS occurs in the ICU most often because critically ill patients are given multiple serotoninergic agents. Many receive opioids, prokinetics and antidepressant medications. They are also routinely prescribed antiemetics, antibiotics and other drugs, where many pose a risk of serotonin release [34].

The main challenge in clinical practice is to recognize or to diagnose SS early on. Many critically ill patients already present an altered mental status—up to 80% of ICU patients are delirious (confusion assessment method, CAM-ICU positive) and show sympathetic hyperactivity (hyperthermia, hypertension, tachycardia, diaphoresis). Moreover, patients often have nausea or diarrhea or suffer from neuromuscular excitation. But these symptoms may be due to a range of diverse causes including hyperthermia, leukocytosis, or delirium in septic patients, tachycardia or hypertension caused by pain, or withdrawal syndrome. Nausea, vomiting, or diarrhea are also caused by enteral nutrition, paralytic ileus and so on.

Several diagnostic criteria for recognizing SS have been published [35,36]. Among them Hunter’s toxicity criteria seem to be the best, showing 84% sensitivity and 97% specificity. The condition is caused by exposure to a serotonin agent and the presence of any of the following: spontaneous clonus, inducible or ocular clonus along with agitation or diaphoresis, inducible or ocular clonus accompanied by increased muscle tone and temperature over 38 °C, or tremor and hyperreflexia [37]. 

The clonus is central to the diagnostic criteria, and it is most commonly elicited by foot dorsiflex, ocular clonus (ping-pong gaze) and spontaneous clonus in the most severe cases. (Table 3).

The activation of 5-HTR1A causes myoclonus, hyperreflexia and changes to the mental status. 5-HTR2A activation is dangerous, is responsible for tachycardia, hypertension, and can lead to renal failure followed by fever and neuromuscular excitation.

The onset of SS ranges from several hours to several weeks following administration of serotoninergic drugs. It typically occurs soon (within 12–24 h) after exposure to medication and resolves within 24 h following discontinuation. This rapid onset and rapid resolution can help in differential diagnosis. Symptoms can be mild to life-threatening. If SS is recognized early, and causative drugs are stopped immediately, the symptoms usually subside. Almost all patients improve with supportive therapy, but it is critical to take SS into account while making a diagnosis. Severe forms can be deadly, mainly through rigidity, extreme hyperthermia, seizures and rhabdomyolysis. Fever is not centrally (hypothalamus) mediated, so antipyretics will not work—physical cooling and nondepolarizing paralysis and intubation are necessary. The basic principle of treatment is to stop causative medications and follow up with high-quality supportive therapy. Only early diagnosis and treatment can prevent complications such as multiple organ dysfunction and death [38,39]. Laboratory evaluations do not help much as lab abnormalities may include leukocytosis, lactic acidosis, disseminated intravascular coagulation (DIC) and rhabdomyolysis with myoglobinuria.

Differential diagnosis can discriminate SS from other toxidromes such as neuroleptic malignant syndrome, malignant hyperthermia, anticholinergic syndrome and sympathomimetic syndrome (see Table 4) [38,40].

More and more patients take antidepressants as a chronic medication. Polypharmacy for chronic pain, common in more than 60% patients with depression, poses a risk of adverse drug interactions. Added to this is the serotoninergic medication in the ICU. Twenty-two different drugs with serotoninergic activity have been identified (See Table 5). 

When ICU patients develop changes in mental status accompanied by signs of neuromuscular excitation, it is crucial to promptly check the list of medications and stop all serotoninergic drugs without delay. When in doubt, drug interaction programs may help.

It is highly challenging for clinicians to identify drug combinations that may increase the risk of SS. Avoiding these drug combinations is often not entirely possible, and selecting the least distressing combinations is therefore crucial. Clinicians must be highly vigilant due to increased polypharmacy of various serotoninergic agents. SSRI are the most common agents associated with SS. Their concurrent use with other serotoninergic drugs increases the risk of SS. 

Once serotoninergic agents are removed, supportive care is the mainstay of treatment. Anxiety, agitation and seizure can be managed with benzodiazepines. Serotonin symptoms can be managed with the 5-HT1A/2A receptor antagonist cyproheptadine, a first-generation antihistamine with a sedative effect and anti-serotoninergic activity. Potential side effects are anticholinergic activity (tachycardia, urinary retention). This agent is available only in the oral form, and its intravenous alternative is chlorpromazine. It is essential to avoid the causes of hyperthermia such as agitation and muscle hypermetabolism. Pain sources in the patient should be treated with appropriate analgesia. The aim is to avoid opioids with serotoninergic potential (fentanyl, oxycodone etc.). Benzodiazepines cause anxiolysis, amnesia, sedation and have an anticonvulsant effect. Nonetheless, benzodiazepines have delirious potential, especially in geriatric patients. This is probably due to enhanced activity of the major inhibitory neurotransmitter gamma-aminobutyric acid, as benzodiazepine acts as a GABA receptor agonist and decreases acetylcholine transmission. Dexmedetomidine, a centrally acting α2 agonist with sympatholytic, sedative, amnestic, anxiolytic and analgesic properties, seems to be advantageous for sedation. It is a highly selective agonist of peripheral α2A and of brain and spinal cord α2B and α2C receptors. There are two mechanisms by which dexmedetomidine may efficiently mitigate SS. Activation of α2C receptors in the striatum modulates serotonin levels, as it appears that α2 receptors located in serotoninergic terminal axons inhibit serotonin release. Stimulation of presynaptic α2A receptors in the prefrontal cortex and locus coeruleus reduces sympathetic tone and causes sedation without respiratory depression [41,42,43].

Thus, a trio of symptoms is typical for SS: changes in mental status, sympathetic hyperactivity and neuromuscular excitation. The medication(s) responsible must be discontinued immediately, and it is then critical to treat agitation to ensure patient comfort and to limit muscle activity that can worsen hyperthermia. Suitable agents include cyproheptadine (sedating antihistamine with anti-serotoninergic activity) and dexmedetomidine (a central α2 agonist), as well as ensuring sufficient analgesia.

## 3. Conclusions

Serotonin regulates a wide range of physiological and pathophysiological processes in most human organs and plays an important role in immunity and inflammation. New serotoninergic drugs open up the possibility of effectively managing a number of diseases. Intensive care specialists must take SS into account, as early diagnosis has the potential to significantly improve a patient’s prognosis.

## Figures and Tables

**Figure 1 ijms-22-04837-f001:**
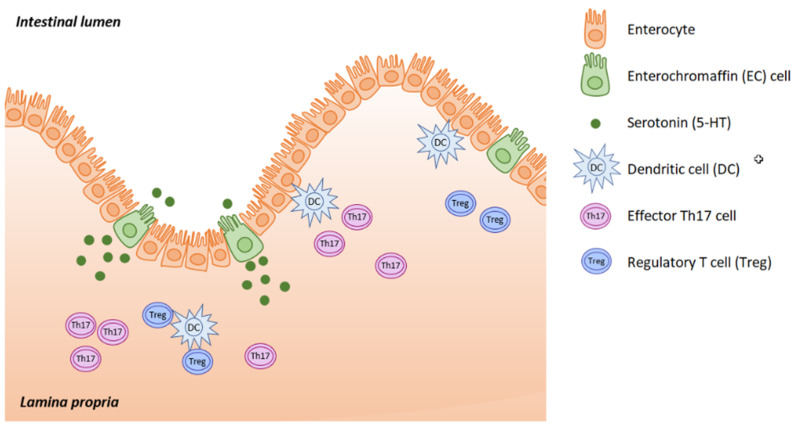
Serotonin is synthesized in EC cells and the majority is released into the lamina propria, with smaller amounts being released into the gut lumen. (Adapted from [1]).

**Figure 2 ijms-22-04837-f002:**
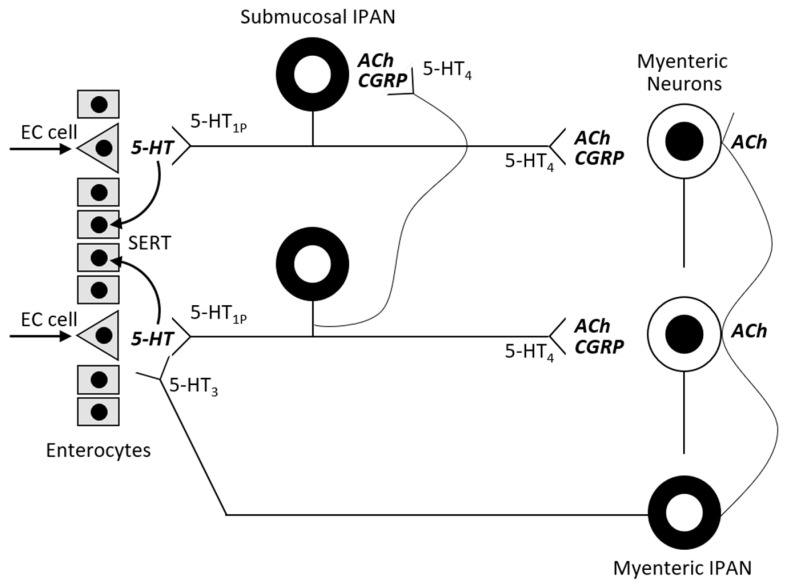
Actions of serotonin in the bowel wall (adapted from [24]).

**Table 1 ijms-22-04837-t001:** Serotoninergic receptors and their distribution.

5-HTR Family	Distribution
5-HTR1	CNS, cardiovascular system
5-HTR2	CNS, PNS, GI tract, platelets, cardiovascular system
5-HTR3	CNS, PNS, GI tract
5-HTR4	CNS, PNS, GI tract, cardiovascular system
5-HTR5	CNS
5-HTR6	CNS
5-HTR7	CNS, PNS, GI tract, cardiovascular system

**Table 2 ijms-22-04837-t002:** Serotoninergic receptors in immune cells.

Monocytes/macrophages	5-HTR 1A, 1E, 2A, 3A, 4, 7
Microglia	5-HTR 2B, 5A, 7
Dendritic cells	5-HTR 1B, 1E, 2A, 2B, 4, 7
Neutrophils	5-HTR 1A, 1B, 2
Basophil, Mast cells	5-HTR 1A
Eosinophils	5-HTR 1A, 1B, 1E, 2A, 2B, 6
B cells	5-HTR 1A, 2A, 3, 7
T cells	5-HTR 1A, 1B, 2A, 2C, 3A, 7
Platelets	5-HTR 2A, 3
NK cells	
Endothelial cells	5-HTR 1B, 2A, 2B, 4
Vascular smooth muscle cells	5-HTR 1D, 2A, 2B, 7

**Table 3 ijms-22-04837-t003:** Hunter’s toxicity criteria.

Mental Status Changes	Anxiety, Delirium
	Seizure, coma
Sympathetic hyperactivity	Hyperthermia
	Hypertension, tachycardia
	Diaphoresis, flushing
	Mydriasis
	Nausea, vomiting, diarrhea
Neuromuscular excitation	Hyperreflexia
	Clonus (spontaneous, inducible) or ocular clonus.
	Tremor
	Rigidity
	Akathisia (inability to stay still)

**Table 4 ijms-22-04837-t004:** Differential diagnosis of serotonin syndrome.

Toxidromes (Main Etiological Factors)	Common Symptoms	Different Symptoms
Anticholinergic syndrome (Dopamine antagonists)	Altered mental status, hot dry skin	Normal reflexes, no clonuses
Neuroleptic malignant syndrome (low dopamine)	Bradykinesia, rigidity, hyperthermia, fluctuating mental status	Symptoms usually develop over several days
Malignant hyperthermia (Inhalation anesthesia, Succinylcholine)	Hyperthermia, hypertonicity	
Sympathomimetic syndrome	Tachycardia, hypertension, altered mental status	No neuromuscular abnormalities, no clonuses

**Table 5 ijms-22-04837-t005:** Mechanisms of serotonin excess.

	Result from Intrasynaptic Serotonin Excess	Drug ^1^
1	Increased synthesis	L-tryptophan
2	Increased release	Amphetamines, Cocaine, Ecstasy, Opioids
3	Decreased reuptake	TCA: Amitriptyline, Imipramine etc., SSRI: Sertraline, Fluoxetine, Citalopram etc., SNRI: Venlaflaxine, Duoxetine Other antidepressants: Trazodone Opioids: Fentanyl, Tramadol etc.,
4	Decreased metabolism CYP inhibitors (CYP3A4, CYP2C19)	MAOI: Moclobemide, Selegiline, Methylene blue Antibiotics/antimycotics: Linezolid, Ciprofloxacin, Fluconazole
5	5-HT receptors agonists	LSD, Triptans, Mirtazapine, Buspirone
6	Increased 5-HT receptor sensitivity	Lithium
7	Others	Antiemetics: Metoclopramide, Ondasetron etc., Antimigraines: Carbamazepine, Triptanes Antiepileptics: Valproate

^1^ SSRI Selective serotonin reuptake inhibitors, SNRI Serotonin noradrenalin reuptake inhibitors, TCA tricyclic antidepressants, MAOI Monoamine oxidase inhibitors. LSD Lysergic acid diethylamide.

## Data Availability

Not applicable.

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
