# Peer review of "Serotonin—Its Synthesis and Roles in the Healthy and the Critically Ill"

_ijms, 2021, doi:10.3390/ijms22094837_

Round 1

Reviewer 1 Report

I made a careful review of the review which is interesting but in my opinion requires appropriate considerations. The title "Serotonin and interventions in intensive care" is not reflected or only partially in the body of the work. In fact, the physiological and physopathological roles of serotonin are described which do not seem to me to justify the reported title. The focus of the review is not well understood if not the roles of serotonin itself. Furthermore, if the goal is to highlight the roles of serotonin I would also mention the role that serotonin has in cancer. Obviously, however, it is necessary to highlight in a clearer form the objective of the review which in my opinion is currently lacking. Based on these considerations, the review must be suitably modified and improved 

Author Response

I am sending an answer to reviewer 1

Reviewer 2 Report

In the manuscript entitled “Serotonin and interventions in intensive care” the authors performed an extensive review on the serotonine (synthesis and roles) in health and disease, summarising the current state of research on this topic and also underlining the importance of serotonine production/accumulation in critically ill patients.

The subject is important and certainly of interest for intensivists and anesthesiologists.

However, there are some issues that must be reconsidered and improved.

Major changes

Title – “Serotonin and interventions in intensive care can be more specific, not  clear enough.

Keywords I suggest serotonine as  the first keyword.

The Introduction section needs to be reorganized, as currently this essential part of the manuscript is missing. The purpose should be to present/introduce the topic, the authors perspective on the topic, and also to contextualize in the broader academic field.

I would suggest that all information (including Figure 1) to be moved on the main text as a first, distinct subsection entitled, for example, “2.1. Serotonine synthesis”.

Main text – In order to increase the clarity and the readability of this manuscript section, I recommend, if it is possible, to insert a final paragraph (conclusion or recommendation) on each subsection.

Minor changes / Specific recommendations

Abstract

Text on lines 13-14 can be rephrased – e.g. This role is very important in intensive care, as in critically ill patients, multiple serotoninergic agents like opioids, antiemetics, antidepressants are frequently used.”

Main text

The information presented on the lines 204-208 were previously discussed on the lines 30-31 and 87-88 and are redundant.

Text on the line 296 –  “The main challenge in  clinical practice is to recognize or to early diagnose SS

Information from Table 3 can be presented in the main text or may be merged with  those listed in Table 4.

Text on the line 321 – “Severe forms can be deadly”, I suggest.

In Table 5 – referring to the Malignant hyperthermia – inhalational anesthesia and succinylcholine are not different symptoms but main etiological factors for this syndrome.  

Text on the line 334 – “More and more patients take antidepressants before hospitalization” I suggest also as a chronic medication.

Text on the line 337 – “When in doubt, drug interaction programs may help (See table 6).” I agree with this statement  but Table 6 actually presents  the main drugs and mechanisms of action. I recommend that Table 6 must be moved  after the text on line 337 “Twenty- two different drugs with serotoninergic activity have been identified”.

Text on the lines 346-47 - Please rephrase, it is not clear; e.g. Clinicians must have an increased awareness …

Text on the lines 349-51 contain information that were already presented.

Author Response

I am sending an answer to reviewer 2

Round 2

Reviewer 1 Report

Dear editor I re-evaluated the review entitled "Serotonin-its synthesis and roles in the healthy and critically ill" and I can point out that the authors answered, albeit partially, to the questions they were asked. I remain critical of the title which, although modified, I still consider it extremely limited compared to what is reported in the body of the manuscript. The text was instead integrated according to the indications that had been suggested. Therefore, while considering it a duty to consider the review now suitable for publication, I remain skeptical about the impact that the publication may have in terms of novelty .... because I believe that the aspect on which the authors work is extremely important and interesting but too limited to develop a review. I therefore believe that a decision on eligibility for publication must be made collegially with the publisher and the other reviewers ...

Reviewer 2 Report

I would like to thank the authors for generating this improved version of the manuscript.

I really appreciate their efforts to write a good and clear introduction and to elaborate conclusive paragraphs for each subchapter of the main text.

The title is now more informative, just a suggestion for clarity “Serotonine – its synthesis and roles in health and critical illness”

As most of the raised issues were solved, I will recommend to accept the manuscript for publication.